# Increased Risk of Tourette Syndrome with Leukotriene Modifier Use in Children with Allergic Diseases and Asthma: A Nationwide Population-Based Study

**DOI:** 10.3390/children9111607

**Published:** 2022-10-22

**Authors:** Min-Lan Tsai, Hsiu-Chen Lin, Chiung-Hui Yen, Jung-Tzu Ku, Shian-Ying Sung, Hsi Chang

**Affiliations:** 1Department of Pediatrics, Taipei Medical University Hospital, Taipei 110, Taiwan; 2Department of Pediatrics, School of Medicine, College of Medicine, Taipei Medical University, Taipei 110, Taiwan; 3Department of Clinical Pathology, Taipei Medical University Hospital, Taipei 110, Taiwan; 4International Ph.D. Program for Translational Science, College of Medical Science and Technology, Taipei Medical University, Taipei 110, Taiwan; 5Office of Human Research, Taipei Medical University, Taipei 110, Taiwan

**Keywords:** leukotriene antagonist, asthma, allergic diseases, Tourette syndrome, epidemiology

## Abstract

(1) Background: Cysteinyl leukotriene receptor antagonists (LTRAs), including montelukast and zafirlukast, are FDA-approved for treating pediatric asthma and allergic diseases. Tourette syndrome (TS), a common neuropsychiatric disorder in children, is associated with allergic diseases and asthma. In this study, we investigated the risk of TS following an LTRA prescription for pediatric allergic diseases. (2) Methods: Children younger than 18 years of age who were newly diagnosed with asthma, allergic rhinitis, or atopic dermatitis between 1 January 2005 and 31 December 2018 and who were registered in the Taiwan National Health Insurance Research Database, which comprises the medical records of nearly 23 million Taiwanese population, were enrolled. LTRA users were matched with randomly selected LTRA non-users by sex, age, asthma-diagnosis year, and urbanization level. In total, 26,984 participants with allergic disease and TS were enrolled and included in the Cox proportional hazards model analysis. (3) Results: Children with allergic disease and asthma treated with LTRAs had a higher risk for TS than LTRA non-users (adjusted hazard ratio 1.376 [95% CI: 1.232–1.536], *p* < 0.001). LTRA users had a significantly higher risk for TS than LTRA non-users with allergic disease. The cumulative incidence of TS was significantly higher in LTRA users than in non-users with allergic diseases and asthma (log-rank test, *p* < 0.0001). (4) Conclusion: A prescription of LTRAs, mainly montelukast, increased the risk of TS among children with asthma, allergic rhinitis, or atopic dermatitis. The mechanism underlying the neuropsychiatric effect of LTRAs needs further investigation.

## 1. Introduction

Tourette syndrome (TS), or Tourette’s disorder, is a developmental neuropsychiatric disorder that is characterized by multiple motor or vocal tics that wax and wane and persist for more than 1 year, according to the Diagnostic and Statistical Manual of Mental Disorders, Fifth Edition (DSM-5) [1,2]. Provisional tic disorder, with either motor and/or vocal tics, are present for less than a year. TS usually occurs in early childhood, and affects 0.56% of the pediatric population in Taiwan [3] and 1–3% in Western countries [4]. In Taiwan, a higher proportion of the pediatric population with TS are male (76%) [5]. Psychiatric disorders, such as attention-deficit hyperactivity disorder (ADHD) and obsessive–compulsive disorder (OCD) frequently co-occur in TS [1]. The association between TS and allergic diseases is well established, based on evidence from population-based studies [6,7,8]. Multiple environmental triggers, such as neuroimmunologic or inflammatory connections, have been postulated as mechanisms that are associated with TS and allergic disease.

Cysteinyl leukotriene receptor antagonists (LTRA), such as montelukast and zafirlukast, are an FDA-approved selective treatment of asthma and allergic disease [9,10]. Montelukast (brand name: Singulair) is widely prescribed in both children and adults for the treatment of both asthma and allergic rhinitis (AR) and is indicated in the prophylaxis and chronic treatment of asthma and allergic diseases in patients aged 2 years and older [11]. Previous guidelines recommend the use of montelukast for the treatment of asthma or other allergic diseases [12,13,14]. Montelukast is the only LTRA that is licensed for use in children younger than 12 years [15]. In general, children tolerate montelukast well, with few side effects, such as gastrointestinal disturbance, respiratory symptoms, skin reactions, and headache [13]. However, recent reports indicate that montelukast potentially confers a higher risk for psychiatric events, such as anxiety, sleep disorder, aggressiveness, and hyperactivity, in children (incidence ≥10%) [16,17,18]. Adult patients treated with LTRA had a 2.26- to 4.5-fold higher risk of allergic granulomatous angiitis in a Dutch database-based analysis [19]. The FDA has issued warnings about a risk of neuropsychiatric events since 2009 [20]. A nested case–control study in 5- to 18-year-old children with asthma in Canada found that new-onset psychiatric events occurred with almost twice the odds ratios in montelukast users compared to other drugs [21].

Although few studies have evaluated the effect of LTRA in neuropsychiatric disorders in children [16,19], the association between LTRA and TS has not been reported. We, therefore, conducted a nationwide population-based study to investigate the effect of LTRA in children with allergic disease in relation to LTRA non-users. We hypothesized that LTRA increases and exacerbates tic behavior in children with allergic disease, including asthma, allergic rhinitis, and atopic dermatitis.

## 2. Materials and Methods

### 2.1. Study Population

This retrospective cohort study used data from the Taiwan National Health Insurance Research Database (NHIRD), which comprises the medical records of nearly 23 million Taiwanese individuals (99.9% of the Taiwanese population) since 1996 [22]. The medical information includes the outpatient and inpatient records, medication prescriptions, and treatment history and can be obtained through a formal request to the Taiwan Health and Welfare Data Science Center (HWDC). The Longitudinal Health Insurance Database (LHID) on a subpopulation of NHIRD comprises 2 million people who were randomly sampled from the 23 million residents of Taiwan. For the protection of patient confidentiality, patient information has been encrypted. Data linkage and processing were only allowed for authorized researchers via a specified computer in a closely monitored room. The study was conducted in accordance with the Declaration of Helsinki and was approved by the Institutional Review Board of Taipei Medical University, Taipei, Taiwan (TMU-JIRB No. 202011088) on 15 January 2021.

### 2.2. Definition

To investigate the association of LTRA and TS, we analyzed the data of individuals younger than 18 years that were obtained from the LHID for the duration between 1 January 2005, and 31 December 2018. Allergic diseases were defined as having at least two episodes in those diagnosed with asthma (ICD-9 code: 493, ICD-10 code: J45), allergic rhinitis (ICD-9 code: 477, ICD-10 code: J30), or atopic dermatitis (ICD-9 code: 691.8, ICD-10 code: L20), and 188,297 potential participants were identified (Figure 1). We divided this population into two groups: the LTRA users as study group 1 and the LTRA non-users as study group 2. The adherence of LTRA use for this cohort was set at ≥30 days. After the exclusion of patients with an LTRA prescription before the diagnosis of allergic diseases and those diagnosed with TS before the index date, there were 26,984 LTRA users and 157,134 LTRA non-users. The index date was the first diagnosis of the allergic disease or the first prescription day of LTRA. We used 1:1 matching by the age and sex of participants in study group 1 in LTRA non-users and obtained study group 2. We defined the comparison group as a 1:1 age- and sex-matched group with the study group 2 in patients without allergic diseases and without LTRA use (Figure 1).

### 2.3. Confounders

ADHD, anxiety disorder, autism, conduct disorder, depression, epilepsy, intellectual disability, learning disorder, OCD, sleep disorder, urbanization, and use medicines, such as inhaled corticosteroids (ICS) or long-acting beta agonists (LABA) for more than 90 days before the index date were considered as potential confounders [2,16,23,24]. We used cluster analysis to stratify the urbanization level into 7 clusters based on consensus data from the year 2000 according to population density, percentage of people with college-level or higher education, percentage of older adults (age > 65 years), percentage of agricultural workers, and number of physicians per 100,000 population [25]. We further collapsed the urbanization level to 3 levels: urban (levels 1–2), suburban (levels 3–4), and rural (levels 5–7).

### 2.4. Outcome Measurement

TS was the main outcome of this study. Patients who had outpatient or inpatient records including ICD-9-CM codes 307.2, 307.3, 333.3, or ICD-10-CM F95 or R25 were defined as having developed TS. Death and loss to follow-up were considered as data censoring.

### 2.5. Statistical Analysis

Continuous data are described as the mean and standard deviation (mean ± SD). Data on demographics and comorbidities were compared between LTRA users and non-users or between LTRA users and the comparison group using the Student’s t-test or chi-square test. The hazard ratios and 95% confidence interval (CI) for TS in LTRA users, non-users with allergic disease, and the comparison group were compared using a Cox proportional hazard model to estimate the risk of TS after adjusting for age, sex, urbanization level, and the presence of various comorbidities as confounders. Kaplan–Meier curves were applied to assess the cumulative incidence curves, and they were examined by the log-rank test. We performed the statistical analysis by the software SAS (version 9.4; SAS Institute, Inc., Cary, NC, USA). The statistical significance level was set at a two-sided *p*-value < 0.05.

## 3. Results

In total, 188,297 children with allergic diseases, including asthma, allergic rhinitis, or atopic dermatitis, were identified from the database between 1 January 2005 and 31 December 2018. We identified 26,984 LTRA users and selected 26,984 children from among 157,134 LTRA non-users as group 2 and 20,461 from patients of non-allergic, LTRA non-users as a comparison group (Figure 1). Among 26,984 children with LTRA users, use for asthma only was 11,285 (42.82%), allergic rhinitis only was 9490 (53.17%), atopic dermatitis only was 2304 (8.54%), and the remainder were the patients with two or more than two allergic diseases who received LTRA (Table 1). In the LTRA users, most children had received 4 or 5 mg montelukast (26,961/26,984 = 99.9%), and children rarely received 20 mg zafirlukast (23/26,984 = 0.085%) daily.

Table 2 lists the demographic characteristics of patients from these three groups. After matching, the distributions of age and sex were similar among groups 1 and 2. The mean age was 7.18 ± 3.79 years, and 58.65% of the individuals were male. LTRA users had a significantly higher percentage of comorbidities including ADHD, autism, epilepsy, intellectual disability, sleep disorder, and ICS/LABA use compared with LTRA non-users. Therefore, we included those factors as confounders. Finally, we included 868 children with TS in this cohort as study group 1 after excluding patients with TS before the index date. We identified 606 and 125 children with TS in the study group 2 and the comparison group, respectively. We estimated the crude HRs and adjusted HRs (aHRs) after adjusting with the confounders for the risk of TS between study group 1 and study group 2, and between study group 2 and the comparison group. During the 13-year follow-up, the adjusted HR of TS for study group 1 compared with study group 2 was 1.376 (95% CI 1.232–1.536, *p* < 0.001), and the adjusted HR of TS for study group 2 compared with the comparison group was 2.962 (95% CI 2.440–3.597, *p* < 0.001) (Table 3). Although patients with allergic diseases who did not receive LTRA had a higher risk of TS, LTRA users had an additional increased risk of TS.

We performed stratified analysis for each group of asthma only or allergic rhinitis only or atopic dermatitis only. The adjusted HRs for TS in LTRA users compared to non-users were 1.484 (95% CI 1.184–1.861, *p* < 0.001), 1.302 (95% CI 1.104–1.535, *p* < 0.01) and 2.350 (95% CI 1.648–3.349, *p* < 0.001) in separate groups of asthma, allergic rhinitis, and atopic dermatitis, respectively, and were all statistically significant (Table 4). We further performed gender-stratified analysis for the risk of TS in the subgroups of children (5–12 years old) and adolescents (12–18 years old) by dividing male children, female children, male adolescents, and female adolescents to understand gender effect for TS between children and adolescents groups. The result demonstrated a significant increase in the adjusted HR for TS in LTRA users compared to non-users in both male (adjusted HR 1.214, CI 1.036–1.423) and female children (adjusted 1.587, CI 1.172–2.150), but not adolescent males and females. Gender did not show a statistically significant increase in HR, but the children group showed a significant increase in HR for TS compared to adolescent groups of LTRA users versus non-users (Appendix A).

Figure 2 shows the Kaplan–Meier curves of the cumulative incidence of TS among the three groups. The incidence of TS was significantly higher in LTRA users than in LTRA non-users among children with asthma, allergic rhinitis, or atopic dermatitis (log-rank test, *p* < 0.0001). However, children with asthma, allergic rhinitis, or atopic dermatitis without LTRA use had a higher significant cumulative incidence of TS compared to children without allergic diseases (log-rank test, *p* < 0.0001).

## 4. Discussion

To the best of our knowledge, this is the first study that used a nationally representative sample and longitudinal dataset to investigate the relationship between LTRA prescription in allergic diseases and risk for TS in children. LTRA, mainly montelukast, was associated with an increased risk of TS in children with allergic diseases including asthma, allergic rhinitis, and atopic dermatitis. LTRA also increased the risk of TS in each allergic disease when we performed the stratified analysis in the subgroups of asthma, allergic rhinitis, and atopic dermatitis. The strengths of this study are its population-based sampling, cross-national design, reduced susceptibility to selection and recall biases, and adjustment for confounders.

TS and tic disorders have been reported in relation to allergic diseases based on the hypothesis of modulation of the immune response through activity- and stress-based alterations of neuromediator systems [26]. Inflammation, microenvironmental change, and immune response are major contributors to asthma, allergic disease, and related neuropsychiatric disorders. Small life events aggravate tic behavior and increase the severity of TS [27]. Whether neuropsychiatric adverse drug reactions (ADRs) such as anxiety or aggravation are induced or further trigger tics or TS events with LTRA use remains unknown.

LTRAs, specifically montelukast, are specific and potent leukotriene receptor inhibitors. Montelukast is one of the most popular long-term medicines that is prescribed in children (<18 years) with asthma and allergic rhinitis in real-world clinical practice [16], although the effectiveness is debatable [14,28]. The FDA reviewed 82 cases of suicide events associated with the use of montelukast and has warned of neuropsychiatric events since 2009 [20]; however, other observational studies found no association between montelukast use and neuropsychiatric events [29,30]. Compared with the pretreatment stage, temperamental behavior, nightmares, and sleep disorders occurred significantly more frequently in both age groups, 3–7 and 8–18 years, with montelukast in an observational study, and quality-of-life scores were significantly affected in both children and parents [31]. Bernard et al. reported an incidence of neuropsychiatric ADRs following montelukast of 16% (95% CI: 10–26), the most frequent of which were irritability, aggressiveness, and sleep disturbance in a nested cohort study [16]. The researchers reported that the relative risk of neuropsychiatric ADRs associated with montelukast versus ICS was 12 (95% CI: 2–90). However, these retrospective studies may have recall bias, selective enrolment, and interview issues. Our population-based study revealed that LTRA, mostly montelukast, was associated with a significantly increased risk of TS with an adjusted hazard ratio of 1.376 (*p* < 0.001).

Children with asthma are frequently prescribed ICS, LABA, or systemic steroids. Either systemic or inhaled steroid use may sometimes affect emotion, mood, or sleep [32]. A study that compared ICS versus montelukast therapy found that the risk of neuropsychiatric ADR-related drug cessation was 12-fold higher in the montelukast treatment group [24]. The severity of asthma or other allergic diseases was related to sleep disorders, anxiety, and depression. In our study, we performed adjustment with ICS and LABA as confounders. Other neuropsychiatric confounders including ADHD, sleep disorder, OCD, headache, epilepsy, and intellectual disability were also adjusted in our study.

The mechanisms mediating LTRA-use-triggered TS are unknown. Cysteinyl-leukotrienes (CysLTs) are well known to have effect on triggering inflammation and bronchoconstriction in asthma patients [33]. Evidence showed that CysLTs exist in the human brain, and montelukast has high brain–blood barrier permeability [34]. It was shown that LTRAs may have an inhibitory effect on neuroinflammatory diseases in older rats, but not in younger rats [35]. Montelukast had a chemopreventive effect in the prevention of various cancers [36]. Recent research showed that montelukast may play a role of modifying viral inflammatory response [37]. A possible explanation of the neuropsychiatric side effect caused by LTRAs in children with allergic diseases may be due to the high blood–brain permeability, thus triggering the occurrence of TS in children.

The pathophysiology of TS and tic disorders have been proposed to emerge from a disruption of the network from the basal ganglia to the prefrontal cortex, i.e., cortico-striato-thalamo-cortical circuits [38]. Multiple neurotransmitter pathways were postulated in relation to TS, specifically dopaminergic, excitatory glutamatergic, and adrenergic pathways [39]. TS symptoms usually wax and wane. Multiple factors contribute to the activation of TS and tics and these include allergic diseases, infectious diseases, psychosocial stress, and epigenetic factors [40]. Children with an underlying vulnerability gene may experience TS symptoms while receiving montelukast therapy. The other reasons may be that children who received montelukast treatment may have had more severe asthma or other allergic diseases, which further trigger psychiatric events such as anxiety, sleep disorders, and TS.

Although the data source is the NHIRD, which provides a large sample of the general population, there were several limitations in our study. First, a significantly higher percentage of confounding factors, including ADHD, anxiety disorder, epilepsy, OCD, sleep disorder, intellectual disability, autism, conduct disorder, and learning disorders, were found in LTRA users compared with non-users in patients with allergic diseases in the multivariate logistic analysis. These results may indicate that children who are LTRA users may have higher rates of psychiatric disorders and may also indicate an interaction among these factors, although we adjusted for these confounding factors. Second, some children were prescribed LTRAs as self-paid medication. This group of patients is missing from our database. Third, the accuracy of diagnostic codes by physicians cannot be absolutely certified, and the regularity of refilled medications cannot fully reflect actual drug compliance. Furthermore, one limitation is that the severity of each of the allergic diseases or comorbidities is not available in the data source of the NHIRD.

In conclusion, this large, nationwide, population-based study revealed that LTRAs, mainly montelukast, increased the risk for TS among children with allergic diseases (asthma, allergic rhinitis, or atopic dermatitis). Further studies are warranted to verify the present findings and investigate the underlying mechanisms of this association.

## Figures and Tables

**Figure 1 children-09-01607-f001:**
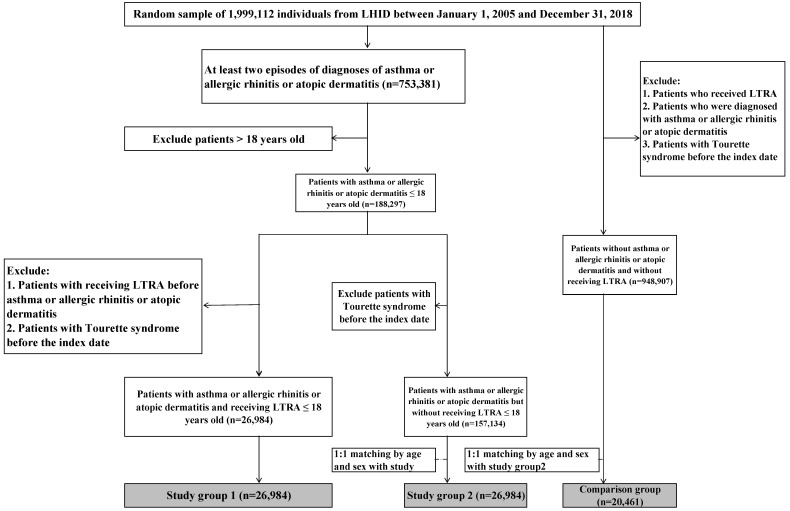
Flowchart of participant recruitment. LHID = Longitudinal Health Insurance Database; LTRA = cysteinyl leukotriene receptor antagonists.

**Figure 2 children-09-01607-f002:**
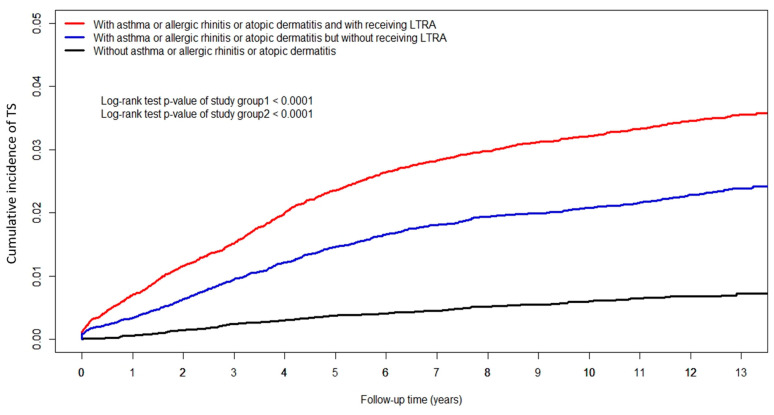
Cumulative incidence of Tourette syndrome (TS). The Kaplan–Meier curves of cumulative incidence of TS among the three groups revealed that occurrence of TS was significantly higher in LTRA users (group 1) than in LTRA non-users (group 2) in children with asthma, allergic rhinitis, or atopic dermatitis (log-rank test, *p* < 0.0001). Children with asthma, allergic rhinitis, or atopic dermatitis without using LTRA also had a higher significant cumulative incidence of TS compared to those without allergic disease (log-rank test, *p* < 0.0001). LTRA = cysteinyl leukotriene receptor antagonists.

**Table 1 children-09-01607-t001:** Percentage of allergic disease in the cohort.

Disease	N	%
Only asthma	11,285	41.8
Only AR	9490	35.2
Only AD	2304	8.5
Asthma and AR	3503	13
Asthma and AD	66	0.3
AR and AD	167	0.6
Asthma and AR and AD	169	0.6
Total	26,984	100

AR: allergic rhinitis; AD: allergic rhinitis.

**Table 2 children-09-01607-t002:** Baseline characteristics, including demographic and other comorbidity factors of patients in study groups 1 and 2 and the comparison group.

Variables	Study Group 1	Study Group 2	Comparison Group		
	Patients with Asthma or Allergic Rhinitis or Atopic Dermatitis		People without Asthma or Allergic Rhinitis or Atopic Dermatitis and without Receiving LTRA (*n* = 20,461)	*p*-Value ^A^	*p*-Value ^B^
	With Receiving LTRA *n* = 26,984	Without Receiving LTRA *n* = 26,984			
Age in years (mean ± SD)	7.18 ± 3.79	7.18 ± 3.79	8.11 ± 3.88	1.0000	<0.0001
Gender, male (*n* (%))	15,825 (58.65)	15,825 (58.65)	11,093 (54.22)	1.0000	<0.0001
ADHD (*n* (%))	911 (3.38)	501 (1.86)	396 (1.94)	<0.0001	0.5329
Anxiety disorder (*n* (%))	277 (1.03)	102 (0.38)	85 (0.42)	<0.0001	0.5194
Autism (*n* (%))	171 (0.63)	99 (0.37)	64 (0.31)	<0.0001	0.3186
Conduct disorder (*n* (%))	17 (0.06)	6 (0.02)	4 (0.02)	0.0218	1.0000
Depression (*n* (%))	19 (0.07)	13 (0.05)	4 (0.02)	0.2887	0.1027
Epilepsy (*n* (%))	645 (2.39)	385 (1.43)	242 (1.18)	<0.0001	0.0212
ICS/LABA (*n* (%))	5815 (21.55)	92 (0.34)	5 (0.02)	<0.0001	<0.0001
Intellectual disability (*n* (%))	196 (0.73)	143 (0.53)	127 (0.62)	0.0039	0.1931
Learning disorder (*n* (%))	71 (0.26)	36 (0.13)	30 (0.15)	0.0007	0.7023
OCD (*n* (%))	35 (0.13)	18 (0.07)	13 (0.06)	0.0195	0.8935
Sleep disorder (*n* (%))	270 (1.00)	114 (0.42)	55 (0.27)	<0.0001	0.0054
Urbanization (*n* (%))	Level 1: 16,392 (60.75) Level 2: 8986 (33.30) Level 3: 1606 (5.95)	Level 1: 16,310 (60.44) Level 2: 8761 (32.47) Level 3: 1913 (7.09)	Level 1: 11,111 (54.30) Level 2: 7579 (37.04) Level 3: 1771 (8.66)	<0.0001	<0.0001

*p*-value ^A^: study group 1 versus study group 2; *p*-value ^B^: study group 1 versus comparison group; ADHD: attention-deficit hyperactivity disorder; ICS: inhaled corticosteroids; LABA: long-acting beta agonists; LTRA: leukotriene receptor antagonists; OCD: obsessive–compulsive disorder; SD: standard deviation. Study group 1: LTRA users; study group 2: LTRA non-users; urbanization 3 levels: urban (levels 1–2), suburban (levels 3–4), and rural (levels 5–7).

**Table 3 children-09-01607-t003:** Crude hazard ratio and adjusted hazard ratio for Tourette syndrome among three groups.

Outcome	Study Group 1	Study Group 2	Comparison Group
	Patients with Asthma or Allergic Rhinitis or Atopic Dermatitis		People without Asthma or Allergic Rhinitis or Atopic Dermatitis and without Receiving LTRA *n* = 20,461
	With Receiving LTRA *n* = 26,984	Without Receiving LTRA *n* = 26,984	
Tourette syndrome (*n*(%))	868 (3.22)	606 (2.25)	125 (0.61)
Crude HR (95%CI)	1.523 (1.373, 1.689) ***	1	
Adjusted HR (95%CI)	1.376 (1.232, 1.536) ***	1	
Crude HR (95%CI)		3.458 (2.852, 4.192) ***	1
Adjusted HR (95%CI)		2.962 (2.440, 3.597) ***	1

Hazard ratios were adjusted for age, gender, urbanization, ADHD (attention-deficit hyperactivity disorder), anxiety disorder, depression, epilepsy, OCD (obsessive–compulsive disorder), sleep disorder, intellectual disability, autism, conduct disorder, learning disorder and ICS (inhaled corticosteroids)/LABA (long-acting beta agonists); LTRA: leukotriene receptor antagonists; CI: confidence interval; HR: hazard ratio; study group 1: LTRA users; study group 2: LTRA non-users; *** *p* < 0.001.

**Table 4 children-09-01607-t004:** Stratified analyses asthma or allergic rhinitis or atopic dermatitis only between LTRA users or non-users.

Outcome	Study Group 1	Study Group 2
	Patients with Asthma or Allergic Rhinitis or Atopic Dermatitis	
	With Receiving LTRA *n* = 26,984	Without Receiving LTRA *n* = 26,984
**Asthma only**	***n* = 11,285**	***n* = 5356**
Tourette syndrome (*n* (%))	329 (2.92)	105 (1.96)
Crude HR (95%CI)	1.557 (1.249, 1.939) *** *p* < 0.0001	1
Adjusted HR (95%CI)	1.484 (1.184, 1.861) *** *p* = 0.0006	1
**Allergic rhinitis only**	***n* = 9490**	***n* = 16,717**
Tourette syndrome (*n* (%))	308 (3.25)	409 (2.45)
Crude HR (95%CI)	1.459 (1.258, 1.692) *** *p* < 0.0001	1
Adjusted HR (95%CI)	1.302 (1.104, 1.535) ** *p* = 0.0017	1
**Atopic dermatitis only**	***n* = 2304**	***n* = 3357**
Tourette syndrome (*n* (%))	88 (3.76)	57 (1.70)
Crude HR (95%CI)	2.655 (1.896, 3.718) *** *p* < 0.0001	1
Adjusted HR (95%CI)	2.350 (1.648, 3.349) *** *p* < 0.0001	1

Hazard ratios were adjusted for age, gender, urbanization, ADHD (attention-deficit hyperactivity disorder), anxiety disorder, depression, epilepsy, OCD (obsessive–compulsive disorder), sleep disorder, intellectual disability, autism, conduct disorder, learning disorder, and ICS (inhaled corticosteroids)/LABA (long-acting beta agonists); LTRA: leukotriene receptor antagonists; CI: confidence interval; HR: hazard ratio; study group 1: LTRA users; study group 2: LTRA non-users; ******* *p* < 0.001, ****** *p* < 0.01.

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
