# Peer review of "Increased Risk of Tourette Syndrome with Leukotriene Modifier Use in Children with Allergic Diseases and Asthma: A Nationwide Population-Based Study"

_children, 2022, doi:10.3390/children9111607_

Round 1

Reviewer 1 Report

Major comments.

Please specify what criteria was used to select the potential confounders included in the study?

Minor comments.

1.     Could stratified analyses be performed for asthma or allergic rhinitis or atopic dermatitis? In case information about severity of each pathology is available, it could be included as confounder to address one of the limitations of the study.

2.     An age-dependent trend for the effect of sex on asthma has been extensively described in the literature. During childhood, females exhibit lower asthma prevalence and incidence, but this trend reverses at adolescence.   Sex-stratified analysis per age group (children vs adolescents) could provide further insights into mechanisms underlying the neuropsychiatric effect of LTRAs.

3.     In Table 2, specify the number of individuals under montelukast and zafirlukast if the information is available. Would the result be consistent for each medication in stratified analysis?

Reviewer 2 Report

Tsai et al. reported “Increased risk of Tourette Syndrome with Leukotriene Modifier use in children with allergic disease and asthma: A nationwide population-based study”. There have been few reports between LTRA and Tourette syndrome. Therefore, basically, this paper is suitable for publication. Some suggested improvements are shown below.

Major points

1, The authors have already mentioned in the background that there is the association between allergic disease and Tourette syndrome (P2 L47-49). Nevertheless, the frequency of psychiatric disorders does not appear to differ between w/o LTRA allergic (Study Group2) and non-allergic disease groups (Comparison Group) in Table 2. In addition, there is a large difference between the LTRA use group (Study Group 1) and the non-use group (Study Group 2), but there is no explanation for these differences.

2, Among allergic diseases, asthma, atopic dermatitis, and allergic rhinitis have different management methods, and the way of using LTRA is also different. It seems easier to understand if the results of sub-analysis by allergic disease are included in supplemental tables or figures.

3, It is considered essential to confirm compliance between the LTRA use group (Study Group 1) and the non-use group (Study Group2). It is necessary to specify how compliance (adherence) was confirmed and how well the patient was able to take the drug before being placed in the Study Group2.

Round 2

Reviewer 2 Report

The authors adequately address what I pointed out. Therefore, this paper is worth accepting